# Scaling Law for Document-Level Neural Machine Translation

**Zhuocheng Zhang[1,2], Shuhao Gu[1,2], Min Zhang[3], Yang Feng[1,2]***

[1]Key Laboratory of Intelligent Information Processing,
Institute of Computing Technology, Chinese Academy of Sciences (ICT/CAS)
[2]University of Chinese Academy of Sciences, Beijing, China
[3]School of Future Science and Engineering, Soochow University, China
zhangzhuocheng20z@ict.ac.cn
shuhaog515@gmail.com
zhangminmt@hotmail.com
fengyang@ict.ac.cn

## Abstract

The scaling laws of language models have played a significant role in advancing large language models. However, the scaling law of document-level neural machine translation remains unclear. In order to promote the development of document-level neural machine translation, we systematically examine the scaling laws in this field. In this paper, we carry out an in-depth analysis of the influence among three factors on translation quality: model scale, data scale, and maximum sequence length. Our results indicate that all three factors have a significant impact on model performance. In particular, increasing the maximum sequence length effectively reduces the context-related errors and improves the overall translation quality. Nevertheless, the sequence length cannot be increased indefinitely, as the number of parameters limits the optimal sequence length. Specifically, we propose a formula describing the empirical scaling law between the model size and the optimal sequence length. Our further analysis shows that the error accumulation problem is the primary factor that hindering further improvement in translation quality for the document-level translation by extending the sequence length.

## 1 Introduction

Neural machine translation (NMT) (Bahdanau et al., 2014) has made great progress in recent years (Barrault et al., 2020; Guo et al., 2022). However, as the input text exceeds a single sentence, sentence-level NMT methods will fail to capture discourse phenomena, such as pronominal anaphora, lexical consistency, and document coherence. In contrast, document-level neural machine translation (DNMT) (Maruf et al., 2021) aims to improve translation consistency and coherence

---

*Corresponding author.

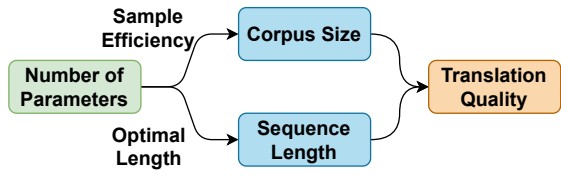

Figure 1: The key factors affecting the translation quality of document-level neural machine translation.

by leveraging contextual information beyond the sentence being translated. Different from previous sentence-by-sentence DNMT methods, which employ an additional context encoder to capture the discourse information (Chen et al., 2021; Miculicich et al., 2018), recent studies have shown that the Transformer (Vaswani et al., 2017) is able to translate multiple sentences directly (Liu et al., 2020; Bao et al., 2021; Sun et al., 2022), referred to as document-by-document translation method. These works demonstrated that increasing the granularity of translation from sentence to document is able to further reduce the context-dependent errors encountered by sentence-level translation methods, such as misinterpretation of pronouns (Müller et al., 2018) and lexical inconsistency (Jiang et al., 2021).

Although has been widely explored, the mechanism of DNMT has not been fully investigated. For example, previous document-by-document DNMT methods are trained on a fixed maximum sequence length, i.e. the document longer than the maximum sequence will be split into smaller segments, and whether further increasing the maximum sequence length can continue to improve the performance of the model still remains to be investigated. Additionally, as the sequence length grows, the difficulty of modeling the contextual information increases further, and whether more training data is needed remains to be explored. Furthermore, the scale of

model parameters is among the most crucial factors affecting the quality of sentence-level NMT methods and is widely studied by the scaling law of neural machine translation (Ghorbani et al., 2021; Ott et al., 2018), but how this factor affects the performance of the DNMT model remains unclear. Finally, it remains an open question whether these factors will interact with each other.

To answer the questions above, we conduct massive experiments, specifically investigating the impacts of three key factors: corpus size, model scale, and maximum sequence length. In addition, we explore the joint effect of these factors. Our findings suggest that all three factors are closely related to the performance of the DNMT model and enlarging the maximum sequence length effectively improves the ability of the DNMT model to capture the contextual information. However, the sequence length can not be increased infinitely as the translation quality begins to drop when the sequence length exceeds a certain limit. Surprisingly, our experimental result shows that this limitation is proportional to the logarithm of the number of model parameters. Our further analysis demonstrates that determining the optimal maximum sequence length requires striking a balance between capturing additional contextual information and managing accumulated errors during auto-regressive decoding. This finding highlights the importance of future research in exploring effective solutions to mitigate this challenge. Furthermore, we find that although the DNMT model is initialized using a pretrained sentence-level translation model, increasing the document-level corpus size can further improve the translation quality. Notably, this improvement increases as the number of model parameters increases. As Figure 1 concludes, our study indicates that the number of model parameters affects the translation quality of DNMT by affecting two key factors: sample efficiency and optimal sequence length. As the number of model parameters increases, both the sample efficiency and the optimal sequence length grow, leading to the performance improvement of the DNMT model.

## 2   Related Work

### 2.1   Document-level Neural Machine Translation

Document-level neural machine translation has witnessed significant advancements in recent years (Maruf et al., 2021). Miculicich et al. (2018) pro-

posed a hierarchical attention mechanism to capture the discourse information while Maruf et al. (2019) employed a selective attention module to select the most relevant information in the context. Lupo et al. (2021) further improved these methods by splitting the sentence into smaller segments to overcome the training sparsity problem. Recently, Tiedemann and Scherrer (2017); Ma et al. (2020) suggest that Transformer has the ability to translate multiple sentences directly, and this document-by-document paradigm further reduce the context related errors. However, the DNMT model may fail to converge if the corpus size is limited and the sequence length is long (Liu et al., 2020). Bao et al. (2021); Zhang et al. (2020) proposed to employ attention masks to prevent the overfitting problem, while Sun et al. (2022) proposed a multi-resolution training strategy to smooth the training difficulty of the DNMT model. However, these methods are all based on a fixed sequence length. Whether further increasing the maximum sequence length can continue to improve the translation quality still remains to be investigated.

### 2.2   Scaling Law

The Scaling law of language models has been widely studied, and have played a significant role in advancing large language models. Kaplan et al. (2020) proposed that the perplexity of the language model is inversely proportional to the logarithm of the size of the training data and the number of the model parameters. Hoffmann et al. (2022) further optimized this scaling law by taking a larger corpus size into consideration. Recently, Touvron et al. (2023) proposed that the inference overhead should also be considered by the scaling law. In the field of neural machine translation, Ott et al. (2018) proposed that enlarging the data size can further improve the translation quality of NMT models, while Ghorbani et al. (2021) further proposed that the cross-entropy loss is also governed by the power law of model size and data size. These works have greatly accelerated the development of large language models and neural machine translation (Wang and Komatsuzaki, 2021; Scao et al., 2022; Costa-jussà et al., 2022). However, the existing scaling law of machine translation is based on the cross-entropy loss of the model, which is not directly related to the translation quality. In addition, the effect of contextual information has not been considered in the scaling law of machine

| Architecture | Params | Params w/o Embedding | MACs / Token | MACs / Token |
|---|---|---|---|---|
| Transformer-small | 48.32M | 31.54M | 48.7M | 67.2M |
| Transformer-base | 60.92M | 44.14M | 61.5M | 79.8M |
| Transformer-big | 209.92M | 176.37M | 210.9M | 247.5M |
| Transformer-base-3 | 38.85M | 22.07M | 39.0M | 48.2M |
| Transformer-base-9 | 82.99M | 66.21M | 83.7M | 111.2M |
| Transformer-base-12 | 105.06M | 88.28M | 105.9M | 142.6M |

Table 1: The statistics of the used model. The first column is the model architecture. The second and the third columns show the total number of parameters and the number of parameters without embedding. The fourth and fifth columns show the number of MACs per token at sequence length equal to 32 and 1024, respectively.

translation. Therefore, it is necessary to explore the scaling law of DNMT.

## 3 Experimental setup

### 3.1 Architecture

We use the Transformer (Vaswani et al., 2017) architecture as our DNMT models. To bring our work closer to cutting-edge research, we leverage several improvements on the standard Transformer architecture, including the pre-norm layer normalization (Xiong et al., 2020) and the Gaussian Error Linear Unit (GELU) activation function (Hendrycks and Gimpel, 2016). To further investigate the impact of different model structures on the quality of document-level translation, we experimented with various model widths and depths. Specifically, the small, base, and big models have 6 layers in both the encoder and decoder. In terms of model width, the small and base models employ an embedding dimension of 512, whereas the large model has an embedding dimension of 1024. Additionally, we employ dimensions of 1024, 2048, and 4096 for the small, base, and big models, respectively, in the feed-forward network. To investigate the model depth on translation quality, we vary the number of layers in the base model from 3 to 12. We use deepspeed [1] to profile the number of parameters and MACs per token for our model. The statistics of our models are shown in Table 1.

### 3.2 Dataset

Compared to sentence-level bilingual corpus, document-level bilingual datasets are limited in size (Chen et al., 2021). To maximize the corpus size, we combine data from 5 high-quality document-level English-German datasets, including EUbookshop (Skadiņš et al., 2014), Open-

Subtitles (Lison and Tiedemann, 2016), Europarl (Koehn, 2005), NewsCommentary, and Tilde-MODEL (Rozis and Skadiņš, 2017), from OPUS[2]. The only pre-processing step we take is tokenizing the dataset using SentencePiece [3] with a joint 32k vocabulary. To perform early stopping and determine the hyperparameters, we randomly extracted 100 documents from the dataset as the validation set. For the test set, we utilize newstest2020 from the WMT competition. The statistics of the combined dataset are shown in Table 2. In order to further investigate the impact of data scale on the performance of DNMT, we randomly sampled subsets of different sizes from the combined dataset to serve as our training sets. The details of the sampled subsets are shown in Appendix A. In our experiments, we segment the document into different lengths, ranging from 32 to 1024 tokens, to investigate the impact of maximum sequence length on the performance of DNMT. The length distribution of our segmented datasets is shown in Appendix B.

| Subset | #Words | #Sents | #Docs |
|---|---|---|---|
| EUbookshop | 172M | 9.3M | 14k |
| OpenSubtitles | 136M | 23M | 29k |
| Europarl | 42M | 1.8M | 10k |
| NewsCommentary | 8.8M | 0.4M | 9.7k |
| TildeMODEL | 23M | 1.6M | 52k |
| Combined | 382M | 36M | 115k |

Table 2: The statistical results of the datasets used in our experiments in terms of word count, sentence count and document count.

---

[1] https://github.com/microsoft/DeepSpeed

[2] https://opus.nlpl.eu

[3] https://github.com/google/sentencepiece

## 3.3 Training

Previous researches indicate that training a document-level model based on a sentence-level model yields better results compared to training a document-level model from scratch (Bao et al., 2021; Miculicich et al., 2018). Therefore, we adopt a two-step training strategy. First, we train a sentence-level model by splitting the documents in the corpus into sentences. Then, we finetune the sentence-level model into a DNMT model using the document-level corpus. During training, we use the Adam optimizer (Kingma and Ba, 2014) with $\beta_1 = 0.9$, $\beta_2 = 0.98$, $\epsilon = 10^{-9}$, and the learning rate is scheduled using the inverse square root method (Vaswani et al., 2017) with warmup steps of 4000. We use the label smoothed cross-entropy loss (Szegedy et al., 2016) with the smoothing value of 0.1. We conduct all experiments on 8 NVIDIA A100 GPUs with a maximum of 4096 tokens per GPU. We use half-precision float (fp16) to accelerate the training process (Ott et al., 2018). The dropout and the attention dropout are tuned for every model from $\{0.1, 0.2, 0.3\}$ and $\{0.0, 0.1, 0.2, 0.3\}$, respectively. We conduct our experiments using the fairseq toolkit (Ott et al., 2019).

## 3.4 Evaluation

In order to make the number of generated sentences matches that of the source document, we add an EOS token at the end of each sentence. We use beam-search with beam size 5 for all our experiments. Following previous study (Liu et al., 2020; Bao et al., 2021), we employ document BLEU (d-BLEU) as our document-level metrics, while sentence BLEU (s-BLEU) as our sentence-level metrics. SacreBLEU[4] (Post, 2018) was used to calculate the above metrics. In addition, we also employ COMET-22 (Rei et al., 2022) and chrF (Popović, 2015) to evaluate the quality of generated sentences. We conduct each experiment three times with different random seeds, and present the average score for analysis.

## 4 Experiments

In this section we present the results of our experiments. First, we analyze the effect of single factor on the translation quality of DNMT in Section 4.1. Then we investigate the joint effect of multiple factors in Section 4.2.

[4]https://github.com/mjpost/sacrebleu

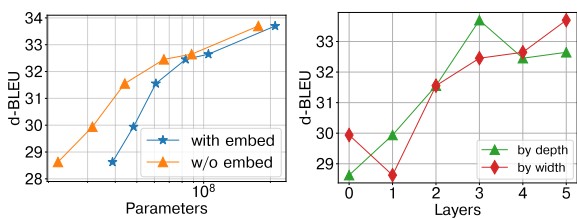

(a) Effect of model parameters    (b) Effect of model shape

Figure 2: The effect of model shape on the performance of DNMT.

## 4.1 Effect of Single Factor

We first investigated the independent impact of two factors, the model and the data, on the performance of DNMT.

**Effect of model size.** To figure out the most important factor affecting translation quality, we train DNMT systems with different model sizes on the combined dataset with a maximum sequence length of 512. As shown in Figure 2, the performance of DNMT increases as the model size increases. However, the d-BLEU score does not increase linearly with the model depth or the model width. Similar to the previous study (Kaplan et al., 2020), observe an approximately linear relationship between the model's performance and the logarithm of the parameter count. Furthermore, this linearity is more closely related to the number of non-embedding parameters rather than the total number of parameters. These findings suggest that compared to the model width and depth, non-embedding parameters better reflects the translation quality of the DNMT model.

**Effect of Data Scale.** To investigate the effect of the data scale on the performance of DNMT, we conduct experiments on different sized subsets of the combined dataset with a maximum sequence length of 512. As shown in Figure 3, the performance of DNMT increases as the data scale increases. Surprisingly, the evaluation shows that regardless of how the model changes, when the number of sentence pairs exceeds 4 million, the performance of the DNMT model will surpass the corresponding sentence-level model. This finding gives us a new understanding of the relationship between data scale and model performance. In addition, the overall trend is consistent with previous studies that there is a power law between model performance and data size (Ghorbani et al., 2021).

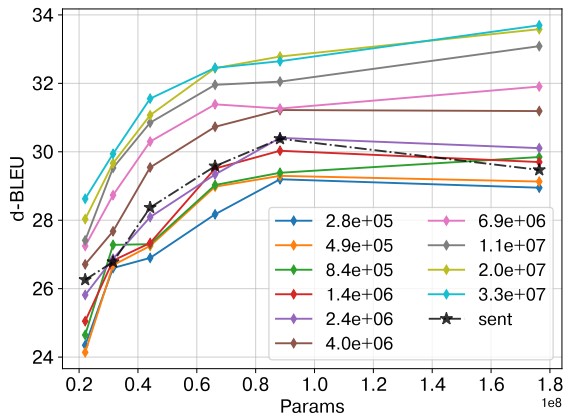

Figure 3: Document BLEU versus the number of parameters. The solid lines represent the performance of document-level models at different data scales, which is measured by the number of sentence pairs, while the black dotted line represents the performance of the sentence-level model.

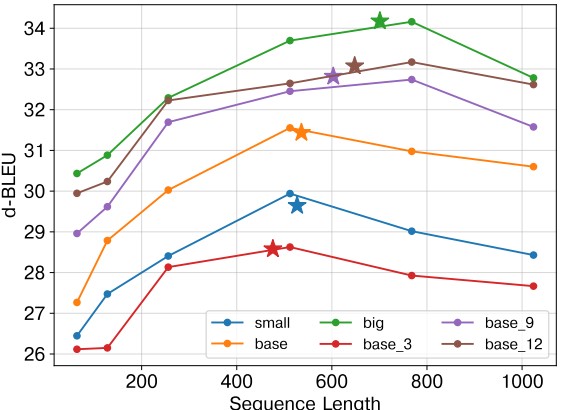

Figure 4: The joint effect of maximum sequence length and model shape on the performance of DNMT. Different colors represent different model shapes, while the star marker represents the estimated optimal sequence length for different model shapes.

## 4.2 Effect of Multiple Factors

To study the interactions between factors, we keep the data scale, model scale, and sequence length fixed separately and investigate the joint impact of the other two factors on translation quality. Specifically, we used the entire dataset to examine the relationship between model scale and maximum sequence length, fixed the model scale as the Transformer-base to study the impact of data scale and maximum sequence length on translation quality, and held the maximum sequence length at 512 to investigate the relationship between model scale and data scale.

**Joint Effect of Maximum Sequence Length and Model Scale.** As shown in Figure 4, the BLEU score generally increases with the sequence length for various model shapes. However, we observe that there is an upper limit to this performance improvement as the length increases. For example, when the sequence length is 512, the BLEU score of the transformer-base is 31.55. However, when the sequence length reaches 1024, the BLEU score decreases to 30.59. This indicates that there exists an optimal maximum sequence length for the DNMT model. In addition, we find that this optimal sequence length is closely related to the model size. To illustrate this relationship more clearly, we estimate the location of the optimal sequence length for different model sizes. As the star marker shown in Figure 4, the optimal sequence length and the model scale are positively correlated. This finding

differs from the previous studies (Press et al., 2021; Beltagy et al., 2020), as they demonstrate that the model's performance consistently increases with an expanding context. We proposed that this is due to the model being influenced by erroneous context in the generation process, i.e. the error accumulation problem, and we further analyze this phenomenon in Section 5.2. In addition, we find that the relationship between optimal sequence length $\mathbf{L}$ and the parameters of the model can be expressed by the following equation:

$$\mathbf{L} = a\log(N) + b. \tag{1}$$

where $N$ is the number of parameters, $a$ and $b$ are constants. We estimate the value of $a$ and $b$ by fitting the data in Figure 4. The estimated value of $a$ is 111, while the estimated value of $b$ is -1399. This indicates that for every increase of 77 tokens in the optimal sequence length, the model parameter size needs to be doubled. This finding is consistent with the previous studies, indicating that a suitable sequence length for the transformer-base model is approximately 512.

**Joint Effect of Maximum Sequence Length and Data Scale.** To figure out whether the data scale affects the ability of the transformer model to handle long text, we conduct experiments on different subsets of the combined dataset with different maximum sequence lengths. As shown in Figure 5, under different data scales, the performance of the model generally increases and then decreases with the increase in sequence length. Similar to our find-

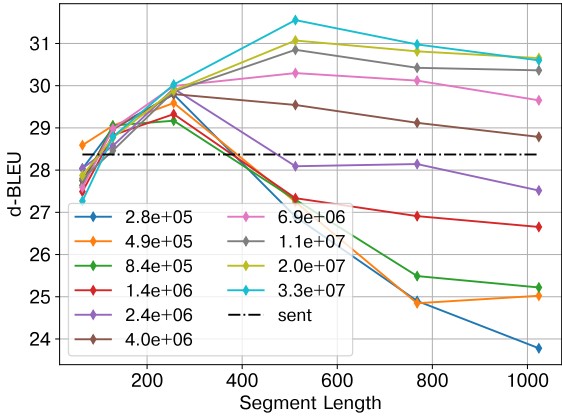

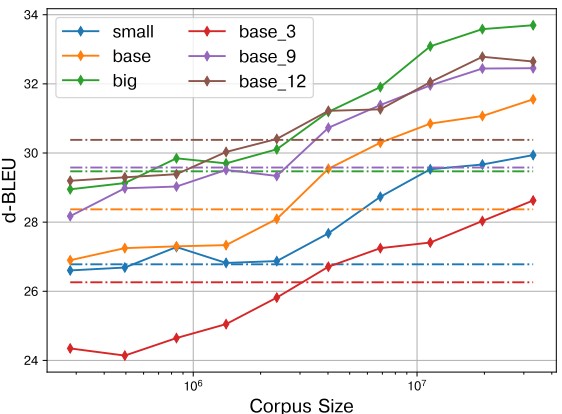

Figure 5: The joint effect of maximum sequence length and data scale on the performance of DNMT. The solid lines represent the performance of document-level models at different maximum sequence length, while the black dotted line represents the performance of the sentence-level model.

Figure 6: The joint effect of data scale and model shape on the performance of DNMT. The solid lines represent the performance of document-level models at different data scales, while the dotted line represents the performance of the sentence-level model.

ings in Section 4.1, when the data scale is large enough, the performance of DNMT will surpass the corresponding sentence-level model. However, although the DNMT models are initialized using a pretrained sentence-level model, the d-BLEU score is lower than the sentence-level NMT model if the data size is limited. In addition, we find that the d-BLEU score is below the sentence-level baseline when the maximum sequence length is 64, even with an adequate amount of data. This is due to the early stopping mechanism terminating the training prematurely when the sequence length is too short. When the maximum sequence length is equal to 128, the BLEU score rises. Surprisingly, the maximum sequence length of 256 is a turning point, where the BLEU score reaches the maximum when the data scale is limited. This indicates that when the document-level corpus size is scarce, we can set the maximum sequence length to 256 in order to maximize the translation quality. In addition, when the maximum sequence length exceeds 512, we observed that the BLEU score shows a declining trend even with further increases in the data scale. This finding further confirms our previous conclusion that there exists an optimal maximum sequence length for a given model.

**Joint Effect of Data Scale and Model Scale.** As shown in Figure 6, we train DNMT models with different data scales and model sizes with a maximum sequence length of 512. The performance of DNMT increases consistently as the data scale increases. In addition, when the number of sen-

tence pairs exceeds 4 million, the performance of the DNMT model outperforms the corresponding sentence-level model, which is consistent with our findings in Section 4.1. In contrast, when the data scale is limited, the models with more layers outperform the models with wider widths. For example, the 12-layer transformer-base model performs better than the transformer-large model with 6 layers when the sentence pairs are less than 4 million, even though the number of parameters in the transformer-large model is twice that of the 12-layer transformer-base model. We believe that this is because when the document-level data is limited, the performance of the DNMT model depends more on the parameter initialization, which is dominated by the sentence-level model. This is consistent with previous studies that sentence-level NMT models are more dependent on the model depth rather than width (Mehta et al., 2020; Wang et al., 2022). However, when the data is sufficient, a model with a larger number of parameters will benefit more than a model with more layers. This indicates that the model's ability to capture context is also related to the model width.

## 5 Discussion

### 5.1 The Correlation Between BLEU Score and Cross-Entropy Loss

As mentioned by Post and Junczys-Dowmunt (2023), The quality assessment of document translation is challenging. In this paper, we demonstrate that the cross-entropy (CE) loss of DNMT model

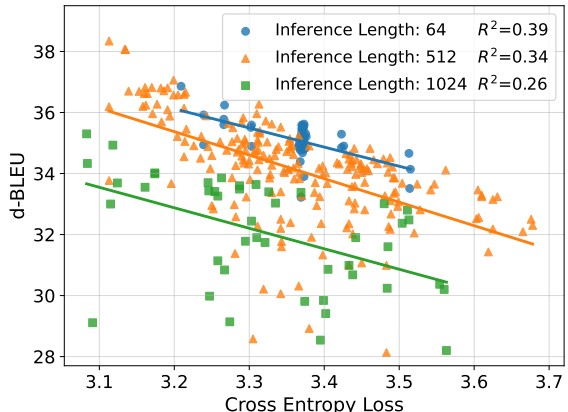

Figure 7: The correlation between cross-entropy loss and d-BLEU score of DNMT model on the validation set. Different colors represent different maximum sequence length, and the solid line represents the linear regression of this correlation. The coefficient of determination are reported in the legend.

does not adequately capture the quality of translation. To further investigate the relationship between the CE loss and the translation quality, we plot the CE loss and the d-BLEU score of DNMT model on the validation set. As shown in Figure 7, the CE loss and the d-BLEU score of DNMT model are weakly correlated, indicating that the loss alone fails to fully depict the translation quality. This finding is inconsistent with the previous study (Ghorbani et al., 2021), which has indicated a substantial association between CE loss and translation quality. In addition, we calculate the coefficient of determination ($R^2$) of the linear regression model. The $R^2$ decreases as the maximum sequence length grows. We suppose that this is because document-by-document DNMT requires generating longer text, which is more prone to the error accumulation problem, which will be discussed in Section 5.2.

## 5.2 The Error Accumulation Problem

To gain a more comprehensive understanding of the underlying reasons for the existence of the optimal maximum sequence length and the failure of CE loss to adequately reflect the translation quality, we analyze the translation quality of sentences in different positions in the documents. We selectively extract a subset from the validation set, where DNMT models under-perform the sentence-level counterparts. Then we employ both sentence-level models and document-level models to translate those documents. We split the translated segments into sentences, and gather them into a subset by there

position in the document. Then we evaluate each subsets via COMET-22 (Rei et al., 2022). The distribution of the number of sentences in different locations is shown in Figure 10. As shown in Figure 8, with the help of contextual information, the translation quality of DNMT model is slightly better than that of the sentence-level model at the beginning of each documents. However, this advantage soon diminishes as the translation proceeds. From the fifth sentence on wards, there has been a growing gap in translation quality between the DNMT model and the sentence-level translation model, which clearly indicates the error accumulation process. This finding demonstrate that the optimal sequence length is actually the trade-off between extra contextual information and the accumulated errors. Interestingly, the gap between the sentence-level model and the document-level model is larger when the model size is smaller, indicating that larger model has a stronger ability to resist the error accumulation problem.

| Context Length | Accuracy |
|---|---|
| 0 | 50.06 |
| 60 | 67.75 |
| 120 | 81.89 |
| 250 | 82.26 |
| 500 | 82.38 |
| 750 | 82.49 |
| 1000 | 81.96 |

Table 3: The accuracy of DNMT model on ContraPro test suite. Context length is the number of tokens in the context. Zero context length indicates that the result is from the sentence-level model.

## 5.3 The Contrastive Experiment

In addition to evaluating the overall translation quality, we also conduct experiments on ContraPro (Müller et al., 2018), a large contrastive test suite extracted from OpenSubtitles 2018 (Lison and Tiedemann, 2016), to evaluate the accuracy of translating the English word "it". As shown in Table 3, all the DNMT models outperform their sentence-level counterparts, clearly indicating that the DNMT model is able to capture contextual information. As the available context increases, the accuracy of the test suite increases in fluctuation. When the context length exceeds 500, the accuracy of the DNMT model stops increasing, and remains at a high level. This tendency is not completely con-

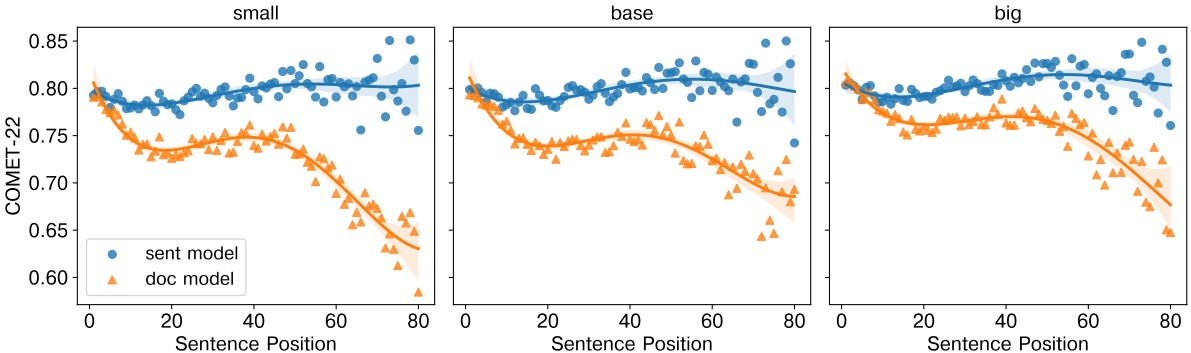

Figure 8: The COMET score of the translated sentences at different positions in the document. The solid lines represent the estimated trend line of the COMET score. As the model size grown, the gap between the DNMT model and the sentence-level model becomes smaller.

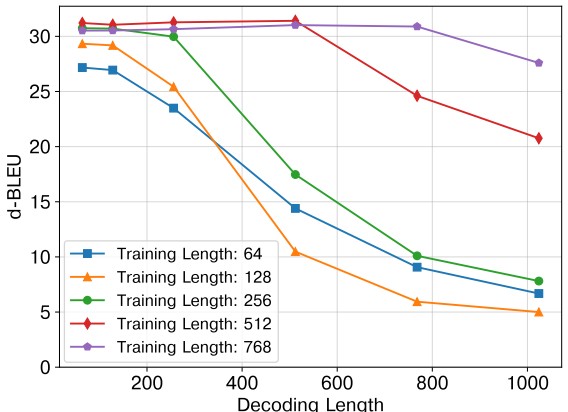

Figure 9: The generalization ability of DNMT model. The models are trained with different maximum sequence length.

sistent with the trend of d-BLEU score, showing that the overall translation quality is also influenced by factors other than the discourse context.

### 5.4 The Generalization Ability

In practice, the lengths of input and output sentences may differ from those encountered during the training phase. Therefore, a proficient DNMT model should be capable of handling documents of various lengths and produce accurate translation results. To investigate this, we segment the documents in the test set into different lengths, and evaluate the DNMT models on the segmented test sets. According to Figure 9, when the sequence length during inference is shorter than that during training, the d-BLEU score of DNMT model decreases mildly. This observation shows that Transformer (Vaswani et al., 2017) is capable of handling documents of various lengths within the maximum

sequence length when finetuned on document-level corpus. Thus it is possible to apply DNMT model to both document-level and sentence-level translation tasks. However, when the inference length exceeds the training length, the performance experiences a sharp drop. We considered that this is because vanilla Transformer lacks length extrapolation ability (Press et al., 2021), which needs to be further investigated in the field of document-level translation.

### 5.5 The computation overhead of DNMT

As the quadratic computational complexity of the attention mechanisms, the computational requirements of the DNMT models are higher than those of sentence-level models. We report the MACs (Multiply–Accumulate Operations) per token required by the DNMT models in Table 1. Interestingly, we find that even though the sequence length increases from 32 to 1024, which is much longer than the optimal sequence length of transformer-big, the MACs per token increase by only 40 percent. This finding indicates that the computation overhead of the DNMT models during inference is not much higher than the sentence-level models, but the errors encounter by sentence-level translation reduced significantly. Thus it is efficient to employ the DNMT model in practice. We have placed a more detailed derivation in Appendix C.

### 6 Conclusion

In this paper, we explore the scaling law of document-level neural machine translation. Our findings suggest that the corpus size, model scale and the maximum sequence length all have significant impacts on the translation quality of the

model. Different from the scaling law of sentence-level neural machine translation, we find that their exists an optimal maximum sequence length for a given DNMT model, and the optimal maximum sequence length is strongly correlated with the number of non-embedding parameters of the DNMT model. We further formulate this correlation with a simple power law. Our additional analysis indicates that the error accumulation problem is the primary factor that hindering further improvement in translation quality for the document-level translation. In future work, we will investigate how to overcome the error accumulation problem, and extend the optimal sequence length of document-by-document translation.

## 7    Acknowledgement

Our research requires a large number of experiments. Thus, we would like to express our gratitude to the ICT computing platform and the technical service team for providing GPU resources.

Furthermore, we thank the anonymous reviewers for their thorough review and valuable feedback.

## Limitations

Although we have conduct massive experiments to explore the scaling law of document-level neural machine translation, there are still some limitations in our work: (1) over four hundred models were trained, resulting in high energy consumption; (2) due to the limitation of the data size, we are unable to conduct a larger scale experiments.

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

## A   The Statistics of Sampled Datasets

To explore the relationship between translation quality and the data size in Section 4, we sampled the combined dataset with different sizes. The size of the subset is determined by the number of documents, and to ensure experiment reproducibility, each size was sampled three times for averaging the results. The statistics of the sampled datasets are shown in Table 4.

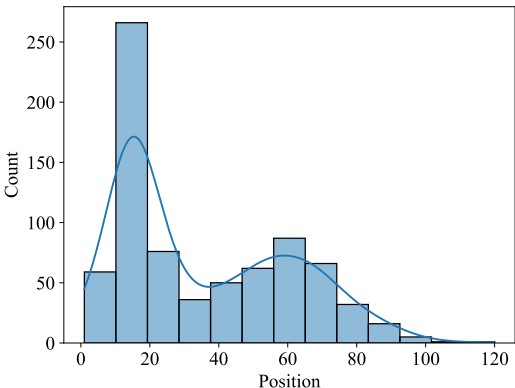

Figure 10: The distribution of the number of sentences in different locations.

## B   The Length Distribution of The Split Dataset

To conduct experiments with different maximum sequence length, we split the documents into segments with different segment lengths. However, due to the document boundary restrictions, we cannot ensure that all segments have the same length. As shown in Figure 11, the length distribution of the training set becomes more even as the maximum sequence length increases, which allows the model to translate the sentences of arbitrary length within the maximum sequence length.

## C   Estimation of Computational Complexity for Transformer Models

To gain a clearer understanding of how sequence length affects the computational cost of the model, we take the Transformer base as an example and provide a detailed derivation of the computational requirements for its forward propagation process. To simplify the derivation, we ignore the computational cost of Layer Norm, residual connection, bias, and the softmax. The embedding dimension of the base model is 512 and the hidden state di-

mension is 512. There are 8 attention heads, and each attention head has a dimension of 64. The feed-forward layer dimension is 2048, and there are 6 encoder layers and 6 decoder layers. We assume that the sequence length of the source and target document is both $n$ and the vocabulary size is 32768. Then we can estimate the computational cost of the model in Table 5.

As shown in the Table 5, the total computational cost of the Transformer base is $n^2 * 18432 + n * 60819456$, and the coefficient of the first-order term regarding $n$ is much larger than the coefficient of the second-order term. Therefore, the growth of computational cost is approximately linear with respect to $n$ when $n$ is small. When substituting sequence lengths $n = 32$ and $n = 1024$ into the above equation, the total computational cost and per-token computational cost are shown in Table 6. As demonstrated in Table 6, although the sequence length has increased significantly, the computational cost per token have not increased significantly. Furthermore, for larger-scale models, due to their expanded hidden state dimensions, the contribution of the linear term will be more significant, which further reduce the impact of sequence length on the computational cost per token.

## D   The Experimental results on Chinese to English dataset.

To further validate our experimental findings, we collected over 10 million parallel sentences from Chinese to English and conducted a repeat of our experiments on this dataset. The composition and scale of this dataset are presented in Table 7. The experimental results, as shown in Figure 12, demonstrate that the overall trends in Chinese-English dataset are consistent with the conclusions we obtained in the English-German dataset. However, there is one significant difference: the optimal sequence length in this dataset is considerably shorter than in the English-German dataset. We believe that the optimal sequence length is also correlated with the data quality, language characteristics, and domain of the dataset. We will leave this issue for future research.

| Subset Number | #Lines | | | #Tokens | | |
|---|---|---|---|---|---|---|
| | 1 | 2 | 3 | 1 | 2 | 3 |
| 1 | 0.28 | 0.29 | 0.27 | 4.29 | 4.27 | 4.08 |
| 2 | 0.52 | 0.49 | 0.48 | 8.13 | 8.05 | 7.62 |
| 3 | 0.82 | 0.86 | 0.85 | 13.12 | 13.92 | 12.85 |
| 4 | 1.39 | 1.42 | 1.40 | 21.77 | 22.28 | 21.48 |
| 5 | 2.37 | 2.41 | 2.30 | 37.70 | 39.21 | 36.30 |
| 6 | 4.00 | 4.06 | 3.99 | 62.64 | 64.93 | 62.83 |
| 7 | 6.81 | 6.88 | 6.87 | 109.17 | 111.99 | 109.54 |
| 8 | 11.47 | 11.39 | 11.51 | 183.84 | 181.83 | 182.84 |
| 9 | 19.53 | 19.66 | 19.58 | 312.61 | 313.62 | 310.31 |

Table 4: The statistics of the sampled datasets in millions. Subset number indicates the number of subsets in the sampled dataset. #Lines and #Tokens indicate the number of lines and tokens in the sampled dataset, respectively.

| Module | Computational Cost | Number of the Module | Corresponding Operation |
|---|---|---|---|
| Embedding | $n * 512$ | 4 | The embedding and positional encoding layers in both encoder and decoder |
| Output Projection | $n * 512 * 32768$ | 1 | The output projection layer for generating the final output probabilities |
| Feed Forward | $n * 512 * 2048 * 2$ | 12 | The feed forward layers in both encoder and decoder layers |
| Attention Project | $n * 512 * 64 * 8 * 4$ | 18 | The query, key, value, and output projection in the multi-head attention |
| Dot Product | $n^2 * 64 * 8 * 2$ | 18 | The dot product between key and query as well as attention weight and value |
| Total | $n^2 * 18432 + n * 60819456$ | 1 | Total computation cost of the Transformer base |

Table 5: The estimation of computational complexity for Transformer base model. We assume that the sequence length of the source and target document is both $n$ and the vocabulary size is 32768.

| $n$ | Computational Cost | Computational Cost / token |
|---|---|---|
| 32 | 1965096960 | 61409280 |
| 1024 | 81606475776 | 79693824 |

Table 6: The computational cost for Transformer base model when $n = 32$ and $n = 1024$.

| Subset | #Words | #Sents | #Docs |
|---|---|---|---|
| UNPC | 417.2M | 17.4M | 94.7k |
| NewsCommentary | 5.3M | 0.1M | 7.8k |
| Combined | 422.5M | 17.5M | 102.5k |

Table 7: The statistical results of the Chinese→English datasets used in our experiments in terms of word count, sentence count and document count.

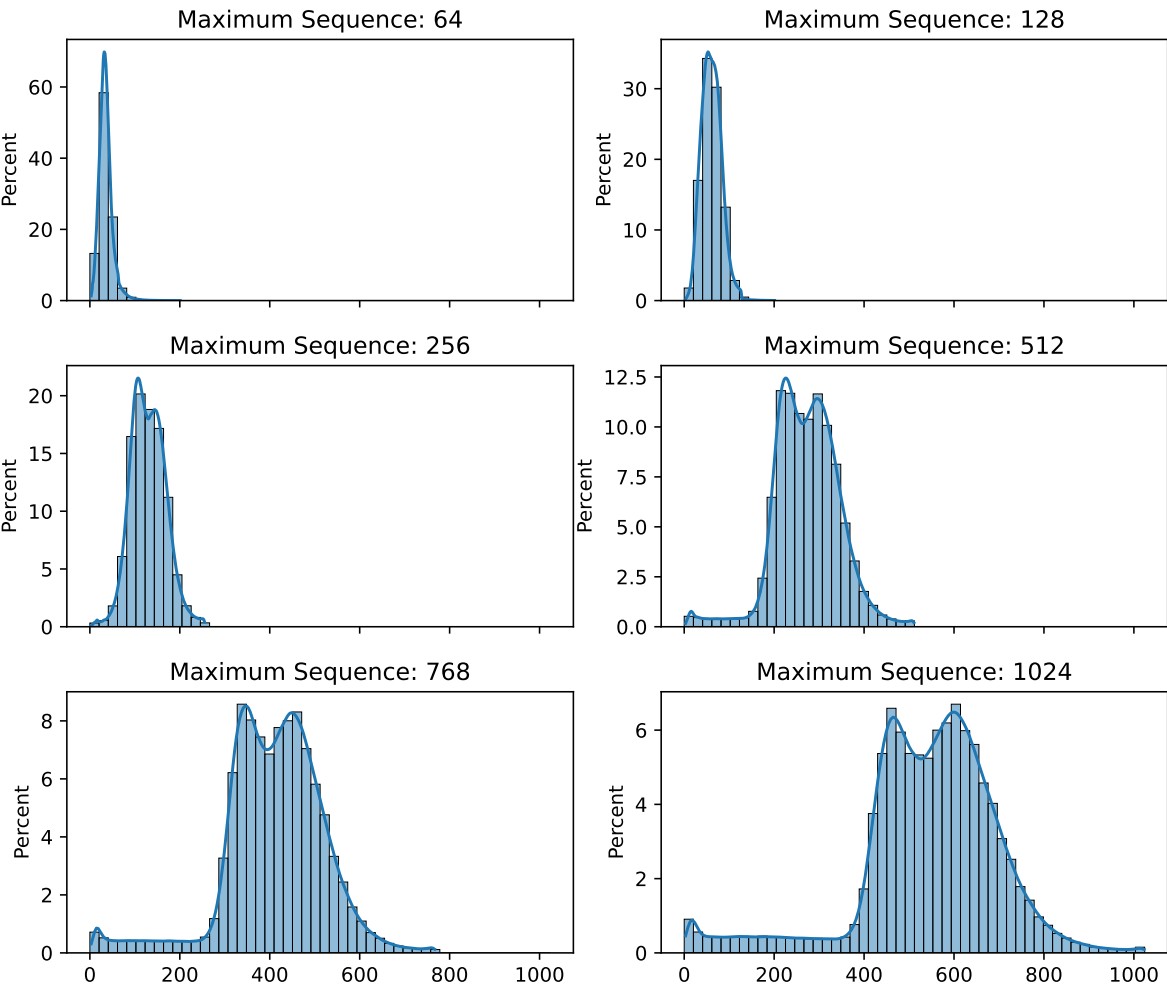

Figure 11: The length distribution of the segmented dataset.

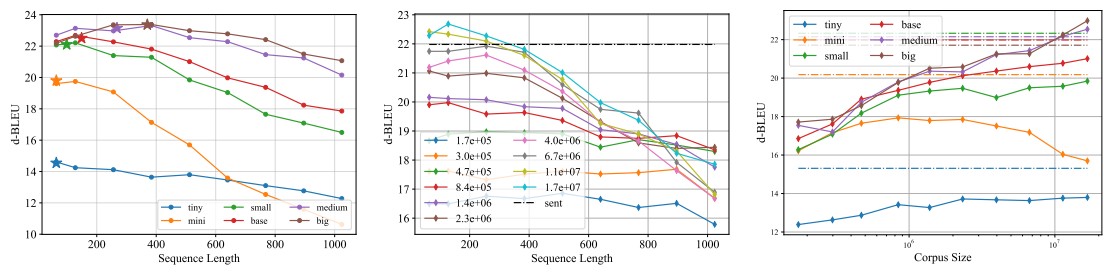

(a) The joint effect of maximum sequence length and model shape.

(b) The joint effect of maximum sequence length and data scale.

(c) The joint effect of data scale and model shape.

Figure 12: The experimental result on Chinese→English dataset.