# OpenReview forum: "Scaling Law for Document Neural Machine Translation"
_EMNLP/2023/Conference — EMNLP 2023 Findings_

### Official Review · Reviewer_zccM · 2023-08-02

**Soundness:** 4

**Excitement:**

3: Ambivalent: It has merits (e.g., it reports state-of-the-art results, the idea is nice), but there are key weaknesses (e.g., it describes incremental work), and it can significantly benefit from another round of revision. However, I won't object to accepting it if my co-reviewers champion it.

**Paper Topic And Main Contributions:**

This paper provides experiments on analyzing the most important three factors in the scaling law of document-level machine translations. Their results show: 1) the number of non-embedding parameters is related to the performance, 2) data size is related to the performance, 3) the optimal sentence length is logarithmically related to the number of parameters 4) the optimal sentence length still exists when increasing the data size 5) model width gets more critical than model depth when the data size gets larger.

They also give further discussions and try to explain these findings in the following sections, by associating it with the error accumulation problem.

**Reasons To Accept:**

Numerous experiments are conducted, and these results are sufficient to support their claims. Interestingly, they even estimated the constant parameters in their Equation (1), thus, providing not only a qualitative but also a quantitative understanding of these phenomena.

**Reasons To Reject:**

I believe most of these findings, as also mentioned by the authors themselves, are not first discovered by them, but are further validated. Therefore, the novelty is considered limited. Besides, the writing needs to be polished.

**Reproducibility:**

5: Could easily reproduce the results.

**Reviewer Confidence:**

3: Pretty sure, but there's a chance I missed something. Although I have a good feel for this area in general, I did not carefully check the paper's details, e.g., the math, experimental design, or novelty.

---

> ### Author Rebuttal · Authors · 2023-08-29
>
> Your feedback is greatly appreciated, and we would like to address your main points of concern to provide a clearer understanding of our work.
>
> ## Q1: The novelty is considered limited.
> This paper provides a systematic investigation of the scaling laws in document-level machine translation. In comparison to the scaling laws observed in sentence-level translation models, we have derived several significant and distinctive findings:
> 1. We have identified the limitations of employing Perplexity (PPL) as an evaluation metric for machine translation tasks, particularly in the document-level machine translation scenario. Our analysis highlights that PPL-based evaluation metrics fail to accurately reflect translation quality. Furthermore, we conducted an in-depth examination of the correlations among different evaluation metrics. The experimental results indicate that PPL gradually deviates significantly from other metrics as sequence length increases;
> 2. Prior work on document-level machine translation has presented two viewpoints. Some studies have indicated that the effective information for improving translation quality is concentrated in the context of approximately 1-3 sentences around the sentence to be translated. On the other hand, another perspective suggests that the model can benefit from contexts of arbitrary lengths. In this paper, we systematically investigate this issue. Through experiments conducted on the ContraPro dataset, we find that the Doc2Doc document-level machine translation model is indeed capable of capturing relevant information from a broader context. **However, when evaluating the model using conventional translation metrics, such as BLEU, we identified an optimal sequence length that corresponds to the best translation quality. Deviating from this optimal length, whether longer or shorter, leads to a degradation in translation quality.** This paper makes the pioneering observation of this phenomenon in document-level machine translation and provides a method, based on model scale, to estimate the optimal context length, which holds significant implications for document-level translation tasks;
> 3. We further explore the underlying reasons behind the optimal context length of document-level machine translation models. It is not the case that more context information is always better, nor is the crucial information solely confined to the preceding 1-3 sentences. Instead, as the available context length increases, the model gains enhanced disambiguation capabilities. **However, due to the limited generation capacity of the model, translation errors inevitably arise, subsequently causing the model to be influenced by erroneous context and thus diminishing the translation quality. This problem is called error accumulation.** As the model scale increases, the translation errors progressively decrease, empowering the model with greater resilience against the accumulation of errors. Consequently, the model can benefit from longer context sequences.
> Based on the above points, we believe that our work is innovative and holds significant significance for the advancement of the field of discourse translation. In addition, we will adjust the structure of the final version of the paper to further highlight our innovation.
>
>
> ## Q2: The writing needs to be polished.
> Thanks for your carefully examination of our submitted paper. We will try our best to enhance the quality and refinement of the writing.
>
> ## Conclusion
> We acknowledge your thoughtful feedback and criticisms of our work. We have carefully reviewed your comments and have provided detailed responses to each of the points. If you find our response satisfactory, we would greatly appreciate an increase in our score. If you have any concerns or doubts about our response, please feel free to contact us during the discussion period.

---

### Official Review · Reviewer_4ECh · 2023-08-03

**Typos Grammar Style And Presentation Improvements:** Statistics in Table 4 is not clear to…
**Soundness:** 4

**Excitement:**

3: Ambivalent: It has merits (e.g., it reports state-of-the-art results, the idea is nice), but there are key weaknesses (e.g., it describes incremental work), and it can significantly benefit from another round of revision. However, I won't object to accepting it if my co-reviewers champion it.

**Paper Topic And Main Contributions:**

The paper examines the impact of scaling model and data sizes and maximum sequence length on document-level neural machine translation. The study is conducted on English to German translation and based on the standard Transformer architecture with improvements. Analysis of the three factors reveled that they are all have impact the translation quality. However, the maximum sequence length is correlated with the number of non-embedding parameters.

**Reasons To Accept:**

Experiments are extensive in all three scaling factors.

**Reasons To Reject:**

The study is conducted on one direction English to German. I am wondering if these conclusions can be extended to other languages.

**Reproducibility:**

4: Could mostly reproduce the results, but there may be some variation because of sample variance or minor variations in their interpretation of the protocol or method.

**Reviewer Confidence:**

4: Quite sure. I tried to check the important points carefully. It's unlikely, though conceivable, that I missed something that should affect my ratings.

---

> ### Author Rebuttal · Authors · 2023-08-29
>
> # Reviewer 2
> Your feedback is greatly appreciated, and we would like to address your main points of concern to provide a clearer understanding of our work.
>
>
> ## Q1: Can these conclusions be extended to other languages?
> Sure! We are conducting new experiments on Chinese-English. However, due to time constraints, we are currently unable to present the experimental results in this round of response. We anticipate being able to complete the experiments in a few days and showcase our results during the discussion phase.
>
> ## Q2: Statistics in Table 4 are not clear.
> Thanks for your careful examination of our submitted paper. We sincerely apologize for the oversight in including units while creating the appendix. In Table 4, the counts for sentences and tokens are both reported in units of millions. We will include this information in the final version of our paper.
>
> ## Conclusion
> We acknowledge your thoughtful feedback and criticisms of our work. We have carefully reviewed your comments and have provided detailed responses to each of the points. If you find our response satisfactory, we would greatly appreciate an increase in our score. If you have any concerns or doubts about our response, please feel free to contact us during the discussion period.

---

### Official Review · Reviewer_38v3 · 2023-08-04

**Soundness:** 3

**Excitement:**

3: Ambivalent: It has merits (e.g., it reports state-of-the-art results, the idea is nice), but there are key weaknesses (e.g., it describes incremental work), and it can significantly benefit from another round of revision. However, I won't object to accepting it if my co-reviewers champion it.

**Missing References:**

A recent paper that highlights important difficulties with document-level MT models (e.g. their evaluation, as mentioned above) is [Post and Junczys-Dowmunt (2023)](https://arxiv.org/pdf/2304.12959.pdf).

**Paper Topic And Main Contributions:**

This paper explores the relationship of the number of parameters, the number of training data points and the maximum sequence length with the performance of transformer-based document-level machine translation models. Additionally, it explores the question why performance of these models drops once the maximum sequence length reaches a critical value.

**Questions For The Authors:**

A: How was the model for the optimal maximum sequence length estimated (Equation 1) and are there any metrics that confirm its reliability?

B: Concerning the analysis in Section 5.2: What maximum sequence length did you use for these experiments? Did you check the number of training examples per sentence position (i.e. how many training examples actually consist of x sentences) and if yes, how does the distribution look like?

C: Concerning the analysis in Section 5.3: Why does the accuracy decrease from context length 60 to 120 and from 120 to 250?

D: I’m surprised about the finding in Section 5.5. What is the setup of this experiment? How did you make sure that the numbers between maximum sequence length 32 and 1024 are comparable?

**Reasons To Accept:**

The paper investigates the maximum sequence length for document-level machine translation, a hyperparameter that isn’t studied well yet even though it is crucial for training performant document-level MT systems in practice.

**Reasons To Reject:**

Based on the analyses presented in the paper, the scaling behavior of document-level machine translation models (in terms of number of parameters and number of data points) is not crucially different from the scaling behavior of sentence-level machine translation systems, which has already been studied by [Ghorbani et al. (2021)](https://arxiv.org/pdf/2109.07740.pdf).

In my opinion, the experiments (or their reporting) could be more thorough in many places to back up the claims made by the authors, e.g.

- Figure 1 indicates a causal relationship between the number of parameters and translation quality by affecting sample efficiency and optimal sequence length. While the graphic is confusing to me (e.g. it looks to me as if the number of parameters is affecting the corpus size), I don’t think that the intended claim (as reiterated at the end of the “Introduction” section) is supported by the results. None of the experiments actually show causation but rather association between the variables.

- There is no information on how the function for the optimal sequence length was estimated (Equation 1) and how reliable we expect this model to be.

- In Section 5.2, the authors should define what they mean by “error accumulation” explicitly. Is the problem that the models base the translation of later sentences made on erroneous translation of earlier sentences? If that is what the authors meant by “error accumulation”, it would need more analysis to verify such a phenomenon. Personally, I’d be interested in details like the maximum sequence length used for these experiments and the number of training examples per bin (which are not reported yet), as the observations made by the authors could be related to tendencies of the Transformer to overfit to length statistics of the training data, see [Varis and Bojar (2021)](https://aclanthology.org/2021.emnlp-main.650.pdf).

- The results presented in Section 5.3 look rather noisy to me as the accuracy on ContraPro decreases from context length 60 to 120 and from 120 to 250 while it improves considerably for larger context lengths. Intuitively, a substantial context length increase (e.g. from 60 to 250 tokens) should not hurt the accuracy. The authors do not make an attempt to explain this trend.

- It would be helpful to also provide the confidence intervals to strengthen the conclusions from the experiments with one or multiple factors.

- The authors mention in Section 5.1 that the cross entropy loss “fails to fully depict the translation quality”. I don’t think that this conclusion is valid based on the authors' experiments. The authors are measuring translation quality in terms of (d-)BLEU, i.e. in terms of a metric that has many limitations, especially for document-level MT, see [Kocmi et al. (2021)](https://aclanthology.org/2021.wmt-1.57.pdf); [Post and Junczys-Dowmunt (2023)](https://arxiv.org/pdf/2304.12959.pdf). If general translation quality is not properly reflected by the metric, then it can’t really be determined whether the loss is a good indicator of general translation quality.

Some minor comments are:
- For most of the experiments in Section 4, the factor that is held constant is not reported, e.g. for the experiment of the joint effect of maximum sequence length and data scale (in Section 4.2), I couldn’t find the model size.
- The authors mention that previous work, in particular, Beltagy et al. (2020) and Press et al. (2021) demonstrate that model performance improves with a larger context. Let me note here that their methods are different from the ones presented in this paper and thus conclusions can be different for a number of reasons.
- In my opinion, a lot of the details on training and inference configurations can be moved to the appendix.
- The abbreviation “MAC” is used in line 194 already but only explained later in the paper.

**Reproducibility:**

3: Could reproduce the results with some difficulty. The settings of parameters are underspecified or subjectively determined; the training/evaluation data are not widely available.

**Reviewer Confidence:**

3: Pretty sure, but there's a chance I missed something. Although I have a good feel for this area in general, I did not carefully check the paper's details, e.g., the math, experimental design, or novelty.

**Typos Grammar Style And Presentation Improvements:**

Line 45: “the” missing
Line 55: “it” missing
Line 168: “necessary” seems quite a strong word
Line 496: Figure 5.3 is mentioned, which doesn’t exist

---

> ### Author Rebuttal · Authors · 2023-08-29
>
> Thanks for your thorough review and constructive feedback. In response to the concern raised about our paper, we summarize our response as follows.
>
> ## Q1: The Scaling behavior of the document-level machine translation system is not crucially different from the sentence-level machine translation system, which has already been studied.
>
> This paper provides a systematic investigation of the scaling laws in document-level machine translation. In comparison to the scaling laws observed in sentence-level translation models, we have derived several significant and distinctive findings:
> 1. We have identified **the limitations of employing Perplexity (PPL) as an evaluation metric for machine translation tasks**, particularly in the document-level machine translation scenario. Our analysis highlights that PPL-based evaluation metrics fail to accurately reflect translation quality. Furthermore, we conducted an in-depth examination of the correlations among different evaluation metrics. The experimental results indicate that PPL gradually deviates significantly from other metrics as sequence length increases;
> 2. Prior work on document-level machine translation has presented two viewpoints. Some studies have indicated that the effective information for improving translation quality is concentrated in the context of approximately 1-3 sentences around the sentence to be translated. On the other hand, another perspective suggests that the model can benefit from contexts of arbitrary lengths. In this paper, we systematically investigate this issue. Through experiments conducted on the ContraPro dataset, we find that the Doc2Doc document-level machine translation model is indeed capable of capturing relevant information from a broader context. **However, when evaluating the model using conventional translation metrics, such as BLEU, we identified an optimal sequence length that corresponds to the best translation quality. Deviating from this optimal length, whether longer or shorter, leads to a degradation in translation quality.** This paper makes the pioneering observation of this phenomenon in document-level machine translation and provides a method, based on model scale, to estimate the optimal context length, which holds significant implications for document-level translation tasks;
> 3. We further explore the underlying reasons behind the optimal context length of document-level machine translation models. It is not the case that more context information is always better, nor is the crucial information solely confined to the preceding 1-3 sentences. Instead, as the available context length increases, the model gains enhanced disambiguation capabilities. **However, due to the limited generation capacity of the model, translation errors inevitably arise, subsequently causing the model to be influenced by erroneous context and thus diminishing the translation quality. This problem is called error accumulation.** As the model scale increases, the translation errors progressively decrease, empowering the model with greater resilience against the accumulation of errors. Consequently, the model can benefit from longer context sequences.
> Based on the above points, we believe that our work is innovative and holds significant significance for the advancement of the field of discourse translation. In addition, we will adjust the structure of the final version of the paper to further highlight our innovation.
>
>
> ## Q2: Figure 1 in the paper is confusing and the casual relationship is not well supported by the experiments.
> Figure 1 illustrates a crucial observation in our experiments, which is that the model scale determines the final translation quality by influencing both sample efficiency and the optimal sequence length. We apologize for any confusion caused by the image in our paper, and we intend to refine this figure in the final version to prevent possible ambiguity.
> In addition, our belief in a causal relationship, rather than a mere correlation, between model scale, optimal sequence length, and sample efficiency is grounded in the following considerations:
> 1. Throughout our experiments, while keeping parameters other than model scale constant, we manipulated the model scale and observed changes in the optimal sequence length and sample efficiency.
> 2. Among model scale, sample efficiency, and optimal sequence length, only the model scale can be predetermined, while the latter two are attributes observable after the model is established.
> Hence, we contend that it is the model scale that determines the optimal sequence length and sample efficiency, rather than solely a correlation.
>
> ## Q3: How the function for the optimal sequence length was estimated (Equation 1) and how reliable we expect this model to be.
> The optimal sequence length used in our experiments is $L=alog(N)+b$, where $a$ and $b$ are parameters to be estimated. This function represents a linear relationship between $L$ and $logN$, thus $a$ and $b$ could be estimated using the least squares method.
> To verify the reliable of our proposed empirical formula, we have conduct experiments from
>
> ## Q4: What is the definition of error accumulation problem?
> In this paper, error accumulation refers to the negative impact on the subsequent text generation due to possible errors in the previous text generation which act as context to the subsequent text generation. Thank you for your suggestion, we will add the definition of error accumulation in the final paper to avoid potential misunderstanding.
>
> ## Q5: What is the maximum sequence length used in Section 5.2 (Error Accumulation Problem)?
> The maximum sequence length is 1024 in this experiment, and the model we use is Transformer base. Thanks for your insightful feedback and we will include this information in our final paper.
>
> ## Q6: What is the length distribution of the dataset (samples per bin) used in Section 5.2?
> In this section, we do not train a new model; instead, we evaluated the performance on a model trained with a maximum sequence length of 1024. During evaluation, we also set the maximum sequence length to 1024. Subsequently, we split the translated segments into sentences and allocated them to different bins based on their positions within the segments. The number of sentences in each bin is illustrated in the table below.
>
> | sentence position | sentence number |
> | ----------------- | --------------- |
> | 0-20              | 12885           |
> | 20-40             | 7088            |
> | 40-60             | 4858            |
> | 60-80             | 1830            |
>
> Once again, we appreciate your valuable feedback, and we apologize for not including this information in the submitted paper. We will incorporate this information in the form of a distribution graph into Figure 8.
>
> ## Q7: The error accumulation may be related to tendencies of the Transformer to overfit to length statistics of the training data.
> It is true that transformer has the tendency of overfitting to length statistics of the training data. However, it is important to note that the maximum sequence length used for both training and testing remains consistent in our experiment in Section 5.2. Therefore, the observed error accumulation problem here is unlikely to be caused by the length bias problem.
>
> ## Q8: Why increasing the maximum sequence length from 60 to 128 hurt the performance on ContraPro?
> Thank you for your careful review of our paper. We also observed this unusual phenomenon during our experiments. We speculate that the fluctuations in accuracy might be attributed to the inherent randomness in the system. To address this, we will repeat these experiments several times and average the results. However, as the limited time, we plan to provide this result during discussion period.
>
> ## Q9: It would be helpful to also provide the confidence intervals to strengthen the conclusions.
> Thank you for your constructive advice. We will incorporate the confidence interval ranges in the final version of the paper. However, due to the complexity of some figures with numerous lines, adding confidence interval ranges to certain figures might potentially impact their readability. As a result, we will carefully consider the trade-off while taking into account the practical aspects.
>
> ## Q10: The BLEU score alone is insufficient to demonstrate that perplexity (PPL) is an inadequate evaluation metrics.
> We greatly appreciate your insights regarding the evaluation metrics. Therefore, we further complement our experiments by calculating the coefficient of determination $R^2$ at different sequence lengths. As shown in the table below, the experimental results demonstrate a significant disparity between PPL and any other translation-specific metrics, confirming that PPL does not effectively reflect translation quality. Therefore, our conclusion remains consistent: PPL is not a truly meaningful measure of translation quality. Due to time constraints, we are unable to include more translation metrics in this analytical experiment. Therefore, we plan to include more evaluation metrics such as COMET and doc-COMET in the final version of our paper.
>
> | $R^2$  | 64   | 512  | 1024 |
> | ------ | ---- | ---- | ---- |
> | d-BLEU | 0.39 | 0.34 | 0.26 |
> | d-chrF | 0.49 | 0.52 | 0.48 |
>
> ## Q11: The factor that is held constant is not reported in Section 4.
> Thank you for your detailed examination of our paper. In Section 4.1, we maintained the model as a Transformer-base as controlled conditions. In Section 4.2, the entirety of the dataset was employed to analyze the joint impact of maximum sequence length and model scale. For the section titled "Joint Effect of Maximum Sequence Length and Data Scale," we again utilized a Transformer-base model. We sincerely appreciate your observation regarding this matter, and in the final version of the paper, we will provide a more explicit description of the variables we have controlled.
>
> ## Q12: The method employed in previous work differs from that used in this paper, thus the conclusion "model performance improves with a larger context" can be different.
> We are grateful to your thoughtful comments on our work. We acknowledge the inaccuracy in the citations here and will make necessary revisions to the final version of our paper.
>
> ## Q13: A lot of the details on training and inference configurations can be moved to the appendix.
> Thank you for your constructive advice. We will further optimize the structure of the paper, highlighting the key points and relocating minor portions to the appendix.
>
> ## Q14: How the MACs per token evaluated?
> We first constructed virtual texts with sequence lengths of 32 or 1024. Subsequently, we set the batch size to 1, and utilizing deepspeed to measured the MACs (Multiply-Accumulate operations) required for both forward and backward passes of a batch. We then divided this value by the sequence length of the text (tokens per batch) to obtain MACs per token. Since this value has been normalized with respect to the number of tokens in each batch, it is comparable between any maximum sequence length.
>
> ## Q15: The grammar errors, typos and not explained abbreviation.
> Thanks for your careful examination of our submitted paper. We apologize for the presence of these errors in the paper and will correct these errors in the final version of our paper. In addition, we will examine the paper more meticulously to prevent such errors.
>
> # Conclusion
> We acknowledge your thoughtful feedback and criticisms of our work. We have carefully reviewed your comments and have provided detailed responses to each of the points. If you find our response satisfactory, we would greatly appreciate an increase in our score. If you have any concerns or doubts about our response, please feel free to contact us during the discussion period.

---

### Meta-Review · Area_Chair_Yra7 · 2023-09-22

**Recommendation:** 4

**Metareview:**

This paper investigates how the number of parameters, the amount of training data, and the maximum sequence length affect the performance of transformer-based document-level machine translation models. It also explores why the performance of these models decreases when the maximum sequence length exceeds a certain threshold.
This paper studies the maximum length of text sequences that can be translated effectively by document-level machine translation (MT) models. This hyperparameter, or tuning knob, is important for training high-performance document-level MT systems in practice, but it has not been well-studied yet. The authors run many experiments that are sufficient to support all claims. However, all experiments are on a single language pair. Nevertheless, the author promised to include Chinese-English results in the camera ready version.

---

### Decision · Program_Chairs · 2023-10-07

**Decision:**

Accept-Findings

**Comment:**

This paper investigates how the number of parameters, the amount of training data, and the maximum sequence length affect the performance of transformer-based document-level machine translation models. It also explores why the performance of these models decreases when the maximum sequence length exceeds a certain threshold.
This paper studies the maximum length of text sequences that can be translated effectively by document-level machine translation (MT) models. This hyperparameter, or tuning knob, is important for training high-performance document-level MT systems in practice, but it has not been well-studied yet. The authors run many experiments that are sufficient to support all claims. However, all experiments are on a single language pair. Nevertheless, the author promised to include Chinese-English results in the camera ready version.